# Cajanin Suppresses Melanin Synthesis through Modulating MITF in Human Melanin-Producing Cells

**DOI:** 10.3390/molecules26196040

**Published:** 2021-10-05

**Authors:** Ponsawan Netcharoensirisuk, Kaoru Umehara, Wanchai De-Eknamkul, Chatchai Chaotham

**Affiliations:** 1Department of Biochemistry and Microbiology, Faculty of Pharmaceutical Sciences, Chulalongkorn University, Bangkok 10330, Thailand; Ponsawan.Ne@student.chula.ac.th; 2Faculty of Pharmaceutical Sciences, Yokohama University of Pharmacy, Kanagawa 2450066, Japan; kaoru.umehara@hamayaku.ac.jp; 3Department of Pharmacognosy and Pharmaceutical Botany, Faculty of Pharmaceutical Sciences, Chulalongkorn University, Bangkok 10330, Thailand; 4Cell-Based Drug and Health Products Development Research Unit, Faculty of Pharmaceutical Sciences, Chulalongkorn University, Bangkok 10330, Thailand

**Keywords:** hyperpigmentation, *Dalbergia parviflora*, tyrosinase, human melanocyte

## Abstract

Despite its classification as a non-life-threatening disease, increased skin pigmentation adversely affects quality of life and leads to loss of self-confidence. Until now, there are no recommended remedies with high efficacy and human safety for hyperpigmentation. This study aimed to investigate anti-melanogenic activity and underlying mechanism of cajanin, an isoflavonoid extracted from *Dalbergia parviflora* Roxb. (Leguminosae) in human melanin-producing cells. Culture with 50 μM cajanin for 48–72 h significantly suppressed proliferation in human melanoma MNT1 cells assessed via MTT viability assay. Interestingly, cajanin also efficiently diminished melanin content in MNT1 cells with the half maximum inhibitory concentration (IC_50_) at 77.47 ± 9.28 μM. Instead of direct inactivating enzymatic function of human tyrosinase, down-regulated mRNA and protein expression levels of MITF and downstream melanogenic enzymes, including tyrosinase, TRP-1 and Dct (TRP-2) were observed in MNT1 cells treated with 50 μM cajanin for 24–72 h. Correspondingly, treatment with cajanin modulated the signaling pathway of CREB and ERK which both regulate MITF expression level. Targeted suppression on MITF-related proteins in human melanin-producing cells strengthens the potential development of cajanin as an effective treatment for human hyperpigmented disorders.

## 1. Introduction

Melanin, a cellular pigment presented in hair, skin and eyes, is generated through melanogenesis pathway in melanocytes that are located at the bottom layer of epidermis [1]. As a photoabsorbent, melanin plays a protective role against skin damage induced by ultraviolet (UV) radiation. However, overproduction and unnecessary accumulation of melanin results in various skin problems, including inflammation, age spots, freckles, and melasma which lessen self-confidence and quality of life [2]. Because of the low efficacy and serious adverse effects associated with available hypopigmentation remedies, the search for a novel, safe and effective anti-melanogenic treatment has been continuously carried on. It has been well accepted that the regulation of tyrosinase, a rate-limiting step enzyme in melanogenesis, is the most effective therapy providing sustained benefits for hyperpigmentation treatment [3]. In recent years, various natural extracts have been introduced as tyrosinase inhibitors which are commercially used in cosmeceutical products as depigmenting agents [4].

In the melanogenesis occurring in cellular melanosome vesicles, the hydroxylation of l-tyrosine amino acid into melanin precursor, l-DOPA and l-dopaquinone is catalyzed by tyrosinase [1,5]. The tyrosinase family proteins consist of tyrosinase, tyrosinase-related protein 1 (TRP-1) and tyrosinase related protein 2 (TRP-2) or dopachrome tautomerase (Dct). These three enzymes control melanogenesis at different steps [6]. Tyrosinase is the major enzyme that generates melanin pigment through conversion of l-tyrosine to l-DOPA and subsequent conversion of l-DOPA to l-dopaquinone. Meanwhile, Dct catalyzes the conversion of dopachrome into 5,6-dihydroxyindole-2-carboxylic acid (DHICA) which is subsequently changed to brown-black melanin (Eumelanin) by TRP-1 [7]. Thus, both up-regulated expression level and activated enzymatic function of tyrosinase family proteins dramatically increase melanin production in melanocytes [8,9]. As a key enzyme accelerating the first and rate-limiting step in melanin biosynthesis, the inhibition of tyrosinase is highlighted as a potential mode to treat hyperpigmentation. Nevertheless, sustainable treatment effect and good safety profile are key requirements for a melanin inhibitor [10]. The expression of tyrosinase is mediated by microphthalmia-associated transcription factor (MITF) which in turn is regulated by cAMP response element (CRE)-binding protein (CREB) [11]. Although the transcription step of MITF is activated by phosphorylated CREB (pCREB), Mitogen-activated protein kinase (MAPKs)/Extracellular signal-regulated protein kinase (ERK) signal promotes MITF degradation [12,13]. Therefore, multi-targeted modulation of MITF has also been proposed as a novel strategy to prolong hypopigmentation effect via suppression on tyrosinase expression [14,15].

Since natural products are recognized as primary sources of therapeutic agents, there has been an increased interest in discovering potential anti-melanogenic compounds in natural extracts [16]. Cajanin, an isoflavonoid isolated from heartwood of *Dalbergia parviflora* Roxb. (Leguminosae) has been previously reported to have potent inhibitory effect against enzymatic activity of tyrosinase isolated from mushroom [17]. However, the effect of cajanin on melanin production in human melanocytes has not been thoroughly examined. This study investigated anti-melanogenic activity and related mechanism of cajanin in human melanin-producing cells. The findings may facilitate the further development of cajanin as an effective depigmenting agent for the treatment of hyperpigmentation disorders.

## 2. Results

### 2.1. Antiproliferative Effect of Cajanin in Human Melanoma MNT1 Cells

Before investigation on anti-melanogenic effect, cytotoxic profile of cajanin (Figure 1) in human melanin-producing cells was primarily investigated. Cell viability was evaluated through MTT assay in human melanoma MNT1 cells (5 × 10^3^ cells/well in 96-well plates) treated with cajanin (0–100 μM) for 72 h. The results showed that treatment with cajanin at 1–50 μM did not cause significant alteration of viability in MNT1 cells compared with non-treated control (Figure 2a).

The cytotoxic effect of 1–50 μM cajanin which did not alter %cell viability after 72-h treatment was further evaluated by a cell proliferation assay. Although 72 h had been reported as a doubling time of human melanocytes [18], the MNT1 cells cultured under the present condition was found to have the doubling time of 13.75 h (Appendix A). Thus, the effect of cajanin on the proliferation of human melanoma MNT1 cells was observed after the incubation period of 24–72 h that covers the logarithmic growth phase of the cells [19]. Figure 2b demonstrates that there was a clear increase in the %proliferation after culturing MNT1 cells at low density (2 × 10^3^ cells/well in 96-well plates) which was not repressed by 0–10 μM of cajanin for 48–72 h. At 50 μM cajanin, on the other hand, the proliferative activity was significantly suppressed compared with the untreated control cells.

### 2.2. Cajanin Diminishes Melanin Content in Human Melanin-Producing Cells

Based on cell viability results, the cajanin range of 1–50 μM which caused no alteration of %cell viabilty after 72-h treatment was selected for further investigation on anti-melanogenic activity. Due to melanin-generating activity, forskolin was used as a positive control [20], while 4-butylresorcinol, a tyrosinase inhibitor, was selected for a negative control [21]. The capability to generate melanin pigment was noted in human melanoma MNT1 cells cultured in DMEM supplemented with 20% FBS, 10% AIM-V medium for 72 h (Figure 3a) and for 24–72 h (Figure 3b). It should be noted that the gradually augmented melanin content in human MNT1 cells was early observed at 12 h of the incubation time (Appendix A). As forskolin is an activator of adenylyl cyclase which consequently stimulates CREB and melanin synthesis, the accumulation of melanin content was obviously observed in MNT1 cells treated with 10 μM forskolin (Figure 3a). The ratio between the cellular melanin and total protein content extracted from MNT1 cells was represented as a relative value to untreated control cells to minimize the interference from antiproliferative effect of cajanin. Figure 3c demonstrates the reduction of cellular melanin content observed after incubation with 10–50 μM cajanin for 72 h. Although the minor diminution of melanin production was presented in MNT1 cells cultured with cajanin at lower concentration (10 μM), the significant anti-melanogenic effect was indicated only in the treatment of 50 μM cajanin. Figure 3b depicts the comparable anti-melanogenesis effect in human melanin-producing cells cultured either with 4-butylresorcinol (20 μM) or cajanin (50 μM) at various time points (24–72 h). Interestingly, cajanin at 50 μM dramatically diminished melanin production in human MNT1 cells early at 24 h of the incubation time (Figure 3d).

The relationship between cellular melanin content and concentration in logarithmic (log) scale was generated for determining the half maximum inhibitory concentration (IC_50_) of cajanin in human melanin-producing cells. As indicated in Figure 3e, the 72-h treatment with cajanin at 77.47 ± 9.28 μM possessed approximately 50% inhibition of melanin production in human MNT1 cells.

### 2.3. Cellular Tyrosinase Activity Suppressed by Cajanin

Since tyrosinase is a key enzyme in melanogenesis, the effect of cajanin on tyrosinase activity was evaluated through both cell-free and cell-based assays. The indirect effect of cajanin on tyrosinase activity was assessed via cell-based model in which the conversion of l-DOPA to dopachrome was catalyzed by tyrosinase contained in the equal amount of cellular protein isolated from cajanin-treated MNT1 cells. Figure 4a demonstrates the significant diminution of tyrosinase activity in the protein lysates prepared from the cells treated with 50 μM cajanin for 24–72 h. It should be noted that the initial reduction of cellular tyrosinase activity observed at 24 h of the incubation time corresponds with the anti-melanogenic effect detected in human melanin-producing cells cultured with 50 μM cajanin. Guided by the previous report of the inhibitory effect on mushroom tyrosinase [17], cajanin at final concentration of 0–50 μM was directly added into the mixture of tyrosinase enzyme extracted from human MNT1 cells and its substrate, l-DOPA, to determine the role of tyrosinase inhibitor. After incubation at 37 °C for 2 h, there was no significant difference of formed dopachrome levels in all testing reactions (Figure 4b). This result indicates that cajanin does not directly inhibit the enzymatic function of human tyrosinase. Notably, mushroom tyrosinase (100 unit/mL) was used as a positive control to confirm to conversion of l-DOPA to dopachrome in cell-free tyrosinase activity assay.

### 2.4. Effects of Cajanin on Melanogenesis-Related Proteins in Human MNT1 Cells

To investigate whether the diminution of cellular melanin and tyrosinase activity in cajanin-treated human melanin-producing cells was associated with the alteration of key melanogenic enzymes, the expression levels of MITF, tyrosinase, TRP-1 and Dct were mainly examined in MNT1 cells cultured with the effective concentration (50 ¼M) of cajanin for 24–72 h through both RT-qPCR and western blot analysis. Figure 5a reveals time-dependent down-regulation of mRNA level of MITF transcription factor in human MNT1 cells cultured with 50 μM cajanin for 24–72 h. Consequently, mRNA levels of MITF-regulated proteins, including tyrosinase (Figure 5b), TRP-1 (Figure 5c) and Dct (Figure 5d) in cajanin-treated cells were significantly suppressed. It is worth noting that the lower mRNA levels of MITF and tyrosinase were sustainably maintained during 24–72 h of the incubation time compared with the non-treated control cells.

Correspondingly, the time-dependent effect of cajanin on protein expression of melanogenesis-related proteins was demonstrated in human MNT1 cells via western blotting. The attenuation of MITF (Figure 6a) and tyrosinase (Figure 6b) protein levels was promptly observed in MNT1 cells cultured with 50 μM cajanin for 24 h. Meanwhile, the significantly decreased expression of TRP-1 (Figure 6c) and Dct (Figure 6d) was indicated after 72 h of cajanin treatment compared with the control group.

### 2.5. Cajanin Modulates MITF Regulating Proteins

The modulatory role of cajanin on the upstream molecules involved in melanogenesis was further elucidated. Since CREB regulatory signals mediate transcription of MITF, the alteration of pCREB/CREB is particularly worthy of investigation. Treatment with 50 μM cajanin significantly diminished pCREB level in human MNT1 cells early at 24 h of the incubation time compared with the control group (Figure 7a). Although, moderated levels of proteins involved with MITF transcription was correspondingly noted in human melanin-producing cells after 24 h of cajanin treatment, there was no alteration of pCREB/CREB at the later times (48–72 h). Thus, the modulation on post-translational step of MITF expression was further investigated in human MNT1 cells cultured with cajanin. Figure 7b indicates that overexpression of pERK, a molecule triggering MITF degradation, was noticeable after incubation of MNT1 cells with 50 μM of cajanin for 72 h. These results suggest that cajanin regulates MITF expression through both pre- and post-translational processes.

## 3. Discussion

It has been documented that a variety of chemical reagents, therapeutic treatments and several pathological conditions are involved with melanogenesis disorders, especially the overproduction of melanin [22]. As MITF stimulates the expression of melanogenic enzymes, including tyrosinase, TRP-1 and Dct, the activation of this transcription factor accordingly increases melanin synthesis in melanocytes [8,23]. Both genetic mutations and exogenous stimuli can alter MITF expression level, consequently leading to melanogenesis dysregulation and pigmentation disorders [24]. Additionally, MITF has been proposed as a diagnostic marker for melanoma [25]. Not only suppression on downstream melanogenic proteins in human melanoma cells but also lightening of skin tone was reported in a clinical study after treatment with a MITF-repressing agent [14]. Interestingly, natural flavonoids have been found to provide an anti-melanogenic effect through down-regulated MITF expression [6]. These findings correlate with the results of this present study wherein the attenuated mRNA (Figure 5a) and protein levels of MITF (Figure 6a) associated with the reduction of cellular melanin content (Figure 3a–e) was revealed in human melanoma MNT1 cells after treatment with 50 μM cajanin, an isoflavonoid isolated from *D. parviflora*, for 24–72 h.

The overproduction of melanin content in various hyperpigmented disorders, including melasma and melanoma is involved with the up-regulation of melanogenesis pathway and hyperproliferation of melanocytes [26,27,28]. Interestingly, the expression of MITF also mediates proliferation and differentiation in melanocytes [29]. It has been reported that natural compound effectively diminished melanin content in melanocytes through suppression of cell proliferation [30]. The antiproliferative effect and down-regulated MITF-related proteins of cajanin (50 μM) strongly support the further development of this isoflavonoid as an effective anti-melanogenic compound.

Cajanin previously showed an inhibitory effect against enzymatic function of mushroom tyrosinase [17]. However, the direct inhibition of cajanin on the activity of tyrosinase isolated from human MNT1 cells was not detected in this study (Figure 4b). Though mushroom tyrosinases are wildly used for high-throughput screening of anti-melanogenic compounds [10], the lack of correlation between the inhibitory effect on mushroom tyrosinase and tyrosinase extracted from human source is also documented [31,32]. The unique structure and amino sequence, including lack of thioether bond closely at the active site and containing EGF domain at the intra-melanosomal region of human tyrosinase cloud be the critical factors involving with the deficient activity against human tyrosinase of various mushroom tyrosinase inhibitors [33,34]. Furthermore, the previous report revealed that only high concentration (200 μM) of cajanin effected 65% and 9.2% inhibition on mushroom and murine tyrosinase activity, respectively [17]. Although cajanin at 50 μM did not act as human tyrosinase inhibitor (Figure 4b), the suppression on MITF-mediated melanogenesis proteins (Figure 5 and Figure 6) strongly support the high efficacy of cajanin for hyperpigmentation treatment in humans.

Human melanoma MNT1 cells are highly pigmented cells that have been widely used for investigating melanogenic activity [19,35]. The increment of cellular melanin content (Figure 3b,d and Appendix A) associating with the sustainable level of melanogenic enzymes, including tyrosinase, TRP-1 and Dct (Figure 6b–d and Appendix A) evidenced the melanogenesis in MNT1 cells during 0–72 h of the incubation time performed in this study. Additionally, MNT1 cells possess the logarithmical growth after culture for 24–72 h (Appendix A) when melanogenic proteins are activated [18]. It should be noted that the gradually decreased MITF expression level after the incubation of MNT1 cells for 24–72 h could be the result of short half-life (approximately 0.5–2.5 h) of MITF [36] and negative feedback mechanism from downstream proteins [37]. Intriguingly, the expression of tyrosinase family proteins in MNT1 cells did not altered correspondingly with the reduction of MITF. It has been reported that tyrosinase is rapidly transferred into melanosome for preventing proteasome degradation and initiating melanin production in hyperpigmented cells [38]. No- tably, the maturation of melanosome from stage I to IV tightly regulates melanogenic ac- tivity of tyrosinase. Despite containing the activity during located in ribosome and rough endoplasmic reticulum, the catalytic function of tyrosinase is completely activated inside melanosome, especially at the latter stages of III and IV [1]. These might involve with the increased activity of tyrosinase observed in MNT1 cells during 6–72 h of the incubation time (Figure 4a and Appendix A). Nevertheless, the prompt diminution at 24 h and sustained decrease of MITF level for 72 h as well as the levels of its downstream melanogenic enzymes, including tyrosinase, TRP-1 and Dct in cajanin-treated MNT1 cells (Figure 6) altogether indicates the prolonged inhibitory effect of cajanin on melanogenesis in human melanin-producing cells.

In terms of targeted modulatory pathway, natural compounds have been reported to attenuate MITF and tyrosinase transcription via restraining cAMP/CREB cascade [39,40]. In this study, the diminished transcription level of MITF in cajanin-treated MNT1 cells (Figure 5a) resulted from the down-regulation of pCREB (Figure 7a). Moreover, cajanin has been indicated to modulate a post-translational process of MITF. It is established that degradation of MITF is stimulated by pERK [41]. Correlating with the modulatory role of various anti-melanogenic natural extracts [42,43], the dramatic decrease of MITF protein level (Figure 6a) is consistent with up-regulated pERK (Figure 7b) observed in MNT1 cells cultured with 50 μM cajanin for 72 h. These results evidently suggested that cajanin might inhibit MITF expression through suppression on gene expression and stimulation of protein degradation consequence with diminished expression of tyrosinase family proteins. Correspondence with the previous report about no significant alteration of MITF during initial period (approximately at 3–24 h) of culturing human melanocytes [44], the modulation of upstream molecules, including pCREB/CREB and MITF were also not observed in MNT1 cells either in the presence or absence of cajanin for 0–12 h (Appendix A).

## 4. Materials and Methods

### 4.1. Chemicals

Cajanin with approximately 98% *w*/*w* was isolated and identified from *D. parviflora* as previously reported [17,45,46]. All chemical reagents, including Dulbecco’s Modified Eagle’s Medium (DMEM) (Cat No. 31053-065), AIM-V medium Albu MAX (Cat No. 3103-5025), trypsin-ethylene diamine tetraacetic acid (Trysin-EDTA; Cat No. 25200-072), phosphate-buffered saline (PBS, pH 7.4; Cat No. 25200-023), fetal bovine serum (FBS; Cat No. 10270-106), antibiotic-antimycotic solution (Cat No. 15240-062), l-glutamine (Cat No. 35050061), 3-(4,5-dimethyl-2-thiazolyl)-2,5-diphenyl tetrazolium bromide (MTT; Cat No. M6494), bicinchoninic acid (BCA) protein assay kit (Cat No. 23225), Luna Universal qPCR Master Mix (Cat No. M3003L), RevertAid First Stand cDNA Synthesis Kit (Cat No. K1621) and DNase I (Cat No. EN0521) were purchased from Thermo Fisher Scientific (Rockford, IL, USA). N, N, N′, N′-tetramethylethylenediamine (TEMED; Cat No. 1610801), 40% acrylamide/bisacrylamide 37.5:1 (Cat No. 161-0148), Tween 20 (Cat No. 1662404) and ammonium persulfate (Cat No. 1610700) were obtained from Bio-Rad (Hercules, CA, USA). Sigma-Aldrich (St. Louis, MO, USA) was a source of radio-immune precipitation assay (RIPA) lysis buffer (Cat No. 20188), protease inhibitor cocktail (Cat No. 04693159001), phosphatase inhibitor cocktail (Cat No. 04906845001), synthetic melanin (Cat No. M8631), bovine serum albumin (BSA) (Cat No. 12659), chloroform (Cat No. C2432), mushroom tyrosinase (E.C. 1.14.18.1; Cat. No. T3824) and 2-propanol (Cat No. I9516). GENEzol reagent (Cat No. GZR100) was purchased from Geneaid Biotech Ltd. (New Taipei City, Taiwan). Antibodies, including mouse monoclonal antibody to MITF (Cat No. ab12039), mouse monoclonal antibody to TRP-1 (Cat No. ab3312), rabbit polyclonal antibody to Dct (Cat No. ab74073) and rabbit polyclonal antibody to β-actin (Cat No. ab8227) were purchased from Abcam (Cambridge, UK). Mouse monoclonal antibody to tyrosinase (Cat No. SC20035) was procured from Santa Cruz Technology (Dallas, TX, USA). Rabbit monoclonal antibody to CREB (Cat No. 9197S), rabbit monoclonal antibody to pCREB (phosphorylated-S133; Cat No. 9198S), rabbit monoclonal antibody to Erk1/2 (Cat No. 4695S), rabbit monoclonal antibody to pErk1/2 (phosphorylated-Thr202/Tyr204; Cat No. 4370S), rabbit monoclonal antibody to GAPDH (Cat No. 2118S), secondary antibody anti-mouse IgG horseradish peroxidase (HRP)-linked (Cat No. 7076S) and anti-rabbit IgG, HRP-linked secondary antibody (Cat No. 7074S) were bought from Cell Signaling Technology (Denver, MA, USA).

### 4.2. Cell Culture

Human melanoma MNT1 cells were purchased from American Type Culture Collection (ATCC, Manassas, VA, USA). The cells were cultured in DMEM containing 20% FBS, 10% AIM-V medium, 1% l-glutamine and 1% antibiotic-antimicrobic under saturated atmosphere of 5% CO_2_ at 37 °C until 70–80% confluence before use in further experiments.

### 4.3. Cell Viability and Proliferation Assay

MNT1 cells at density of 5 × 10^3^ cells/well in 96-well plates were cultured in complete DMEM containing various concentrations (0–100 μM) of cajanin for 72 h then further incubated with MTT solution (0.5 mg/mL) for 3 h at 37 °C protected from light. After removing MTT solution, DMSO was added to dissolve the formed formazan crystals. The absorbance of purple formazan color was measured via microplate reader (Anthros, Durham, NC, USA) at 570 nm. Percent (%) cell viability was representative of optical density (OD) ratio between treated to non-treated control cells. Antiproliferative effect of cajanin was additionally determined to generate cytotoxic profile in human melanin-producing cells. MTT viability assay was performed after culture MNT1 cells (2 × 10^3^ cells/well in 96-well plates) with cajanin for 24, 48 and 72 h. Percent cell proliferation represents the relative OD ratio between the treatment at each time point to the non-treated control cells at 24 h.

### 4.4. Determination of Cellular Melanin Content

Human MNT1 cells at density of 1 × 10^5^ cells/well were seeded into 6-well plates for attachment overnight. After treatment with cajanin (0–50 μM) for 24–72 h, the cells were washed with PBS and lysed with RIPA buffer containing 1% protease inhibitor and 1% phosphatase inhibitor for 45 min at 4 °C. Melanin-containing pellets were collected after centrifugation at 12,000 rpm for 10 min and further dissolved in 200 μL of 1 N NaOH/10% DMSO at 80 °C for 3 h. The absorbance value of the obtained melanin was measured at 405 nm and converted to melanin content compared with standard curve of synthetic melanin. The melanin content was normalized to the total amount of cellular protein and represented as a relative value to untreated control cells.

### 4.5. Cell-Free Tyrosinase Activity Assay

Direct inhibition on tyrosinase activity was assessed as previously described [47]. MNT1 cells were cultured until 70–80% confluence then collected by trypsinization, and centrifugation at 5000 rpm for 5 min (4 °C). The cell pellets were lysed with 1% Triton X-100 in PBS containing 1% protease inhibitor and 1% phosphatase inhibitor at 4 °C with continuous vortexing every 10 min for 1 h. Then, the cell lysate was centrifuged at 12,000 rpm for 15 min (4 °C) to collect the supernatant. Cajanin at final concentrations of 0, 5, 10 and 50 μM in PBS (pH 6.8) was added into the supernatant composing of equal amount of 100 μg protein. The reaction solution was then incubated at 37 °C for 10 min before adding 2 mM of l-DOPA and further incubation for another 2 h at 37 °C. The OD of the forming dopachrome was measured via a microplate reader at 490 nm. The direct effect of cajanin on tyrosinase activity was determined via the following formula:$$\mathrm{Tyrosinase}\;\mathrm{activity}\;\left(\%\right)=\frac{\mathrm{OD}490_{\mathrm{Cajanin}\;(0–50\;\mu M)}}{\mathrm{OD}490_{\mathrm{Control}\;(\mathrm{PBS})}}\times 100$$

### 4.6. Evaluation on Cellular Tyrosinase Activity

The enzymatic determination of tyrosinase isolated from cajanin-treated human melanin-producing cells was modified from Lv et al., 2020 [30]. Human MNT1 cells in 6 well-plates (1 × 10^5^ cells/well) were cultured in DMEM containing cajanin (0–50 μM) for 24–72 h. Then, the cells were collected by trypsinization and centrifuged at 5000 rpm for 5 min (4 °C). The cell pellets were mixed with 1% Triton X-100 in PBS (1% protease inhibitor and 1% phosphatase inhibitor) by using vortex every 10 min for 45 min at 4 °C. After centrifugation at 12,000 rpm for 15 min (4 °C), the supernatant was collected and determined for total protein content by BCA assay kit. The equal amount of 100 μg protein of each sample was mixed with 2 mM of l-DOPA (50 μL) and further incubated at 37 °C for 2 h. The absorbance of the formed dopachrome was measured by a microplate reader at 490 nm. Tyrosinase activity in each treatment was calculated relatively to the non-treated control cells at 24 h according to the formula mentioned above.

### 4.7. Western Blot Analysis

MNT1 cells (1 × 10^5^ cells/well) in 6-well plates were cultured in complete DMEM with or without 50 μM cajanin for 24–72 h. The cells were harvested and lysed with RIPA buffer containing 1% protease inhibitor and 1% phosphatase inhibitor for 45 min on ice. Then, the equal protein samples determined by BCA assay kit were mixed with loading buffer, heated at 95 °C for 5 min and further subjected to 10% sodium dodecyl sulfate-polyacrylamide gels (SDS-PAGE). The separated proteins in polyacrylamide gels were transferred onto nitrocellulose membranes and blocked with 5% skim milk in TBST (25 mmol/L Tris-HCl; pH 7.4, 125 mmol/L NaCl, 0.1% Tween 20) at room temperature for 1 h. The membranes were incubated overnight with specific primary antibody at 4 °C then washed 3 times × 7 min with TBST. Horseradish peroxidase (HRP)-conjugated secondary antibody was added onto the membrane to interact with respective primary antibody for 2 h at room temperature. The signal from the target protein was visualized by using an enhanced chemiluminescence HRP substrate (Thermo Scientific, Rockford, IL, USA). The intensity of protein signal was quantified using analyst/PC densitometric software (Bio-Rad, Hercules, CA, USA).

### 4.8. Determination of mRNA Expression Level by RT-qPCR

Human MNT1 cells (1 × 10^5^ cells/well) in 6 well-plates were cultured with 0–50 μM of cajanin for 24–72 h. Total RNA content was isolated from the treated cells using GENEzolTM reagent according to the manufacturer’s instruction. cDNA was synthesized using RevertAid Premium Reverse Transcriptase. The obtained cDNA was amplified by specific forward and reverse primers as following:-Human GAPDH forward primer: 5′-GAGTCCACTGGCGTCTTCA-3′-Human GAPDH reverses primer: 5′-TTCAGCTCAGGGATGACCTT-3′-Human MITF forward primer: 5′-TCATCCAAAGATCTGGGCTATGACT-3′-Human MITF reverse primer: 5′-GTGACGACACAGCAAGCTCAC-3′-Human tyrosinase forward primer: 5′-TCATCCAAAGATCTGGGCTATGACT-3′-Human tyrosinase reverse primer: 5′-GTGACGACACAGCAAGCTCAC-3′-Human TRP-1 forward primer: 5′-AAGGCTACAACAAAAATCACCAT-3′-Human TRP-1 reverse primer: 5′-ATTGAGAGGCAGGGAAACAC-3′-Human Dct forward primer: 5′-GCAGCAAGAGATACACAGAAGAA-3′-Human Dct reverse primer: 5′-TCCTTTATTGTCAGCGTCAGA-3′

Reverse transcription quantitative real-time PCR (RT-qPCR) was performed with a C100 Thermal Cycler (Bio-Rad CFX384 real-time pcr system) using Luna Universal qPCR Master Mix (M3003). Reaction combinations were incubated for 39 cycles of 95 °C for 5 s; 60 °C for 30 s and 65 °C for 5 s. The target mRNA expression was determined as the relative comparison, which was established with the ΔCt method, comparative threshold (Ct) of target genes, which were normalized with GAPDH (Ct) values.

### 4.9. Statistical Analysis

All data were obtained from three individual experiments and presented as mean ± standard error of mean (SEM). The differences between means of the individual groups were analyzed using one-way analysis of variance (ANOVA) via GraphPad Prism 9.0 software (San Diego, CA, USA). *p*-value < 0.05 was considered as statistical significance.

## 5. Conclusions

Cajanin inhibits melanin synthesis in human melanin-producing cells via suppression on MITF-related proteins (Figure 8). Down-regulated mRNA and protein levels of MITF, which is sequentially mediated by dysregulation on pCREB/CREB and pERK/ERK signals is induced by cajanin (50 μM). Moreover, the reduction of melanin synthesis enzymes, including tyrosinase, TRP-1 and Dct (TRP-2) as well as cellular melanin content is presented in cajanin-treated human melanocytes. The data obtained from this study would strengthen the potential development of cajanin as an effective depigmenting agent for the treatment of hyperpigmentation disorders.

## Figures and Tables

**Figure 1 molecules-26-06040-f001:**
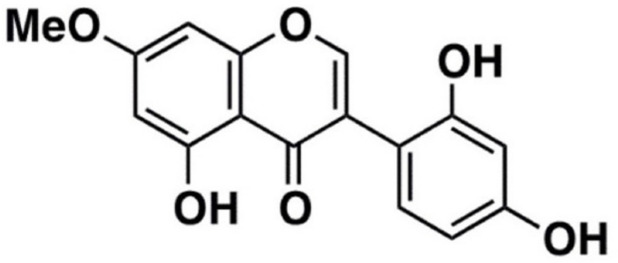
Schematic chemical structure of cajanin.

**Figure 2 molecules-26-06040-f002:**
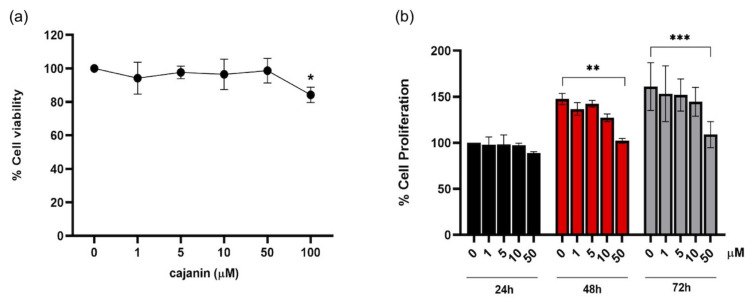
Cytotoxic profile of cajanin in human melanin-producing cells. (**a**) The significant reduction of %cell viability determined by MTT assay was noted in human melanoma MNT1 cells treated with 100 μM cajanin for 72 h. Although there was no alteration of viability observed with the lower concentrations (1–50 μM), (**b**) the incubation with 50 μM cajanin for 48–72 h obviously suppressed proliferation in MNT1 cells. Experiments were performed in triplicate and data are expressed as mean ± SEM. * *p* < 0.05, ** *p* < 0.01, *** *p* < 0.005 versus non-treated cells.

**Figure 3 molecules-26-06040-f003:**
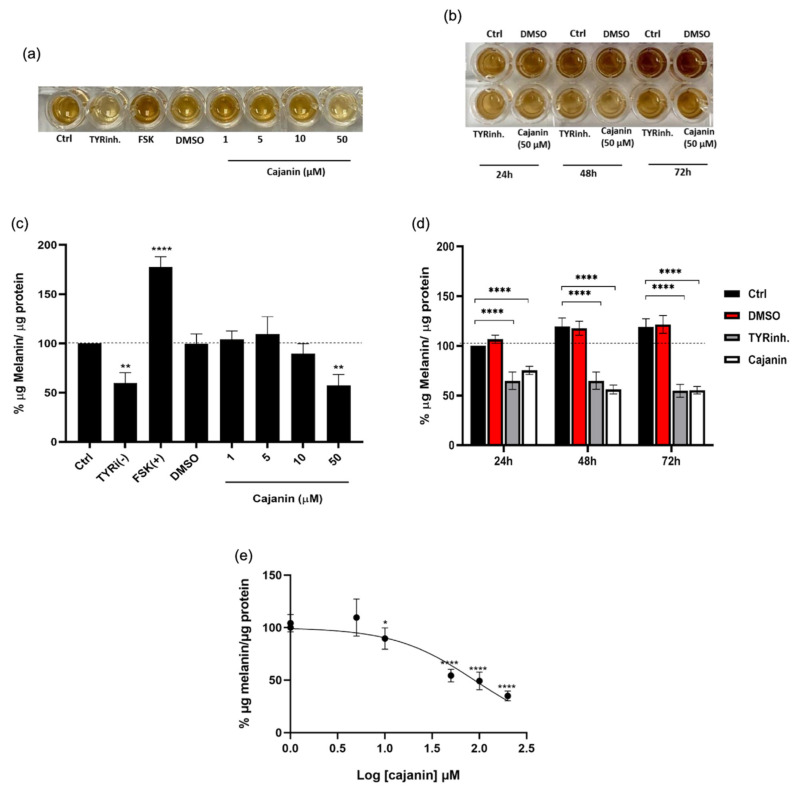
Cajanin inhibits melanin production in human MNT1 cells. (**a**,**c**) After treatment with cajanin (1–50 μM), 20 μM tyrosinase inhibitor (TYRi; 4-butylresorcinol), 10 μM forskolin (FSK) or vehicle (DMSO) for 72 h, the cellular melanin content was determined in human melanoma MNT1 cells. (**b**,**d**) The inhibitory effect of cajanin (50 μM) on melanin synthesis in MNT1 cells was promptly observed at 24 h and sustained until 48–72 h of the incubation time. (**e**) The dose response curve demonstrates 50% reduction of melanin content in human melanin-producing MNT1 cells cultured with cajanin approximately at 77.47 ± 9.28 μM for 72 h. The melanin content is calculated from the ratio between amount of cellular melanin and total protein content relatively to the untreated control cells. Experiments were performed in triplicate and data are expressed as mean ± SEM. * *p* < 0.05, ** *p* < 0.01, **** *p* < 0.001 versus non-treated cells (Ctrl).

**Figure 4 molecules-26-06040-f004:**
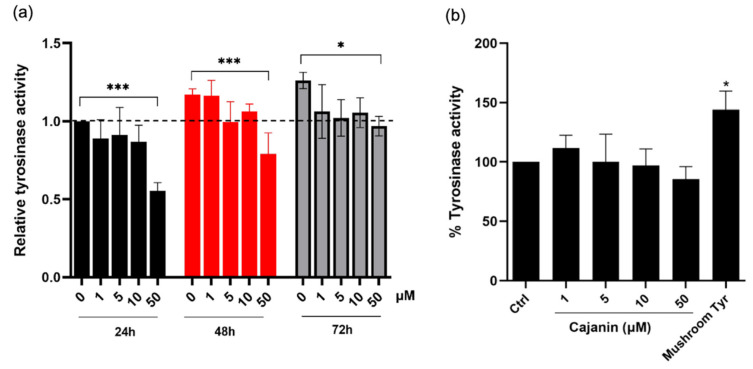
Inhibitory effect of cajanin on human tyrosinase activity assessed via cell-based and direct enzymatic assays. (**a**) Cellular protein lysates prepared from cajanin-treated human MNT1 cells showed lower tyrosinase activity. The significant attenuation of tyrosinase activity was observed in MNT1 cells cultured with 50 μM cajanin for 24–72 h compared with the untreated control group. (**b**) Direct adding of cajanin at different concentrations (1–50 μM) did not alter the melanogenic activity of tyrosinase enzyme isolated from human melanin-producing MNT1 cells. Mushroom tyrosinase (Tyr) at 100 unit/mL was used as a positive control for cell-free tyrosinase activity assay. Experiments were performed in triplicate and data are expressed as mean ± SEM. * *p* < 0.05, *** *p* < 0.005 versus non-treated cells (Ctrl).

**Figure 5 molecules-26-06040-f005:**
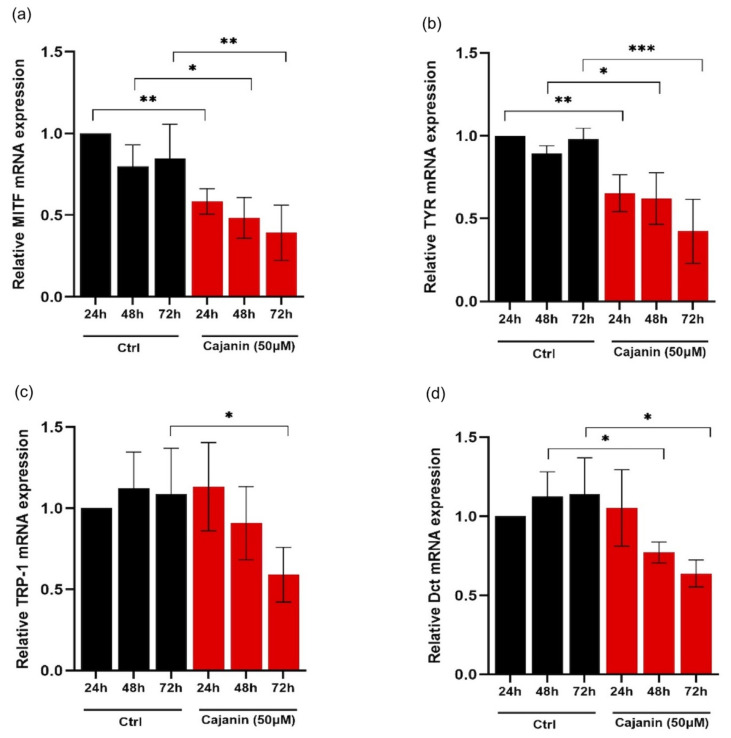
Real-time quantitative RT-PCR reveals time-dependent down-regulation of the mRNA levels of (**a**) MITF, (**b**) tyrosinase (TYR), (**c**) TRP-1 and (**d**) Dct in human melanin-producing MNT1 cells after incubation with 50 μM cajanin for 24–72 h. Data obtained from qRT-PCR were normalized to GAPDH expression level and represented relatively to control at 24 h. Experiments were performed in triplicate and data are expressed as mean ± SEM. * *p* < 0.05, ** *p* < 0.01, *** *p* < 0.005 versus non-treated cells (Ctrl).

**Figure 6 molecules-26-06040-f006:**
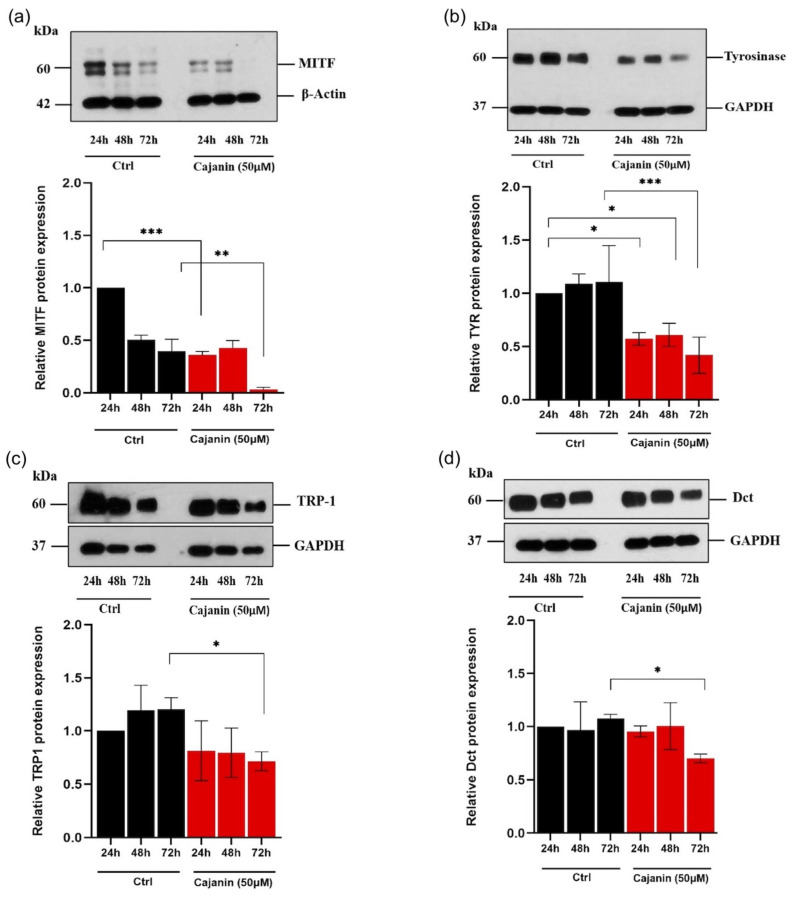
Cajanin attenuates expression levels of melanogenesis-related proteins in human MNT1 cells. After treatment with 50 μM of cajanin, western blot analysis depicts the reduction of (**a**) MITF and (**b**) tyrosinase (TYR) protein levels early at 24 h of the incubation time. Meanwhile, the protein expression levels of (**c**) TRP-1 and (**d**) Dct were significantly diminished in MNT1 cells treated with cajanin (50 μM) for 72 h compared with the control cells. All experiments were performed in triplicate and data are expressed as the mean ± SEM. * *p* < 0.05, ** *p* < 0.01, *** *p* < 0.005 versus non-treated cells (Ctrl).

**Figure 7 molecules-26-06040-f007:**
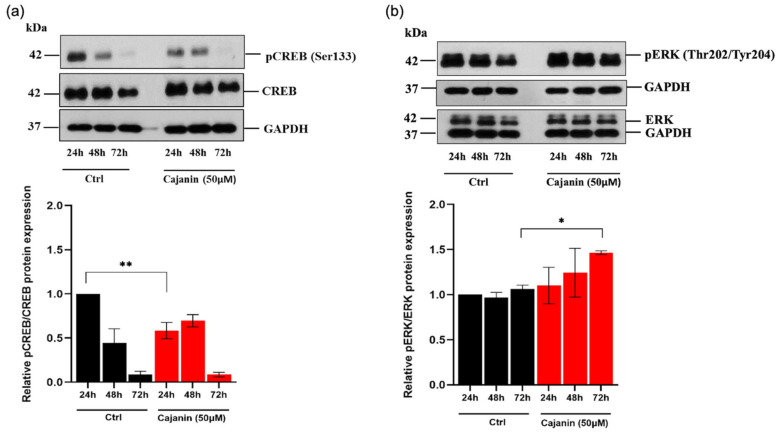
Cajanin regulates signaling molecules mediating MITF expression. (**a**) Suppression on expression level of pCREB/CREB, an upstream signal stimulating MITF transcription was remarkably demonstrated via western blotting in human melanin-producing MNT1 cells cultured with 50 μM cajanin early at 24 h. (**b**) Treatment with cajanin (50 μM) for 72 h also obviously augmented level of pERK which moderates MITF expression at a post-translational step. All experiments were performed in triplicate and data are expressed as the mean ± SEM. * *p* < 0.05, ** *p* < 0.01 versus non-treated cells (Ctrl).

**Figure 8 molecules-26-06040-f008:**
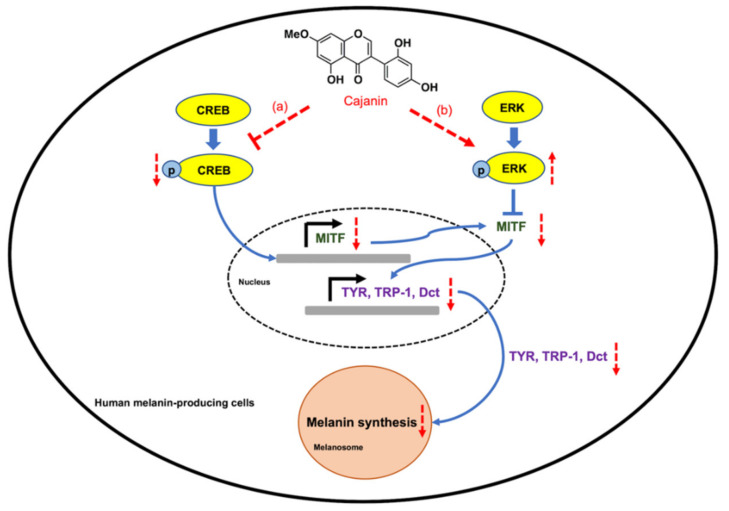
Schematic diagram presenting a proposed anti-melanogenic activity of cajanin in human melanin-producing cells. Cajanin modulates MITF, a melanogenic transcription factor, at both pre- and post-translational steps through down-regulation of pCREB/CREB and activation of pERK/ERK signal, respectively. (**a**) At the initial period of treatment (24–48 h), the inhibition of pCREB/CREB signaling corresponds with decreased mRNA and protein levels of MITF and tyrosinase (TYR) mediated by cajanin. (**b**) Interestingly, the activation of pERK/ERK might play a role on dramatic decrease of MITF protein which consequently attenuates the expression levels of melanin synthesis enzymes, including tyrosinase (TYR), TRP-1 and Dct (TRP-2) at the latter time point (72 h). These modulations on melanin-regulating proteins eventually result in the prolong decreased melanin content in cajanin-treated melanocytes. The red arrows and red symbols indicate the effect of cajanin in human melanin-producing cells.

## Data Availability

Not applicable.

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
