# Peer review of "Cajanin Suppresses Melanin Synthesis through Modulating MITF in Human Melanin-Producing Cells"

_molecules, 2021, doi:10.3390/molecules26196040_

Round 1
Reviewer 1 Report
The authors investigated the molecular mechanism of cajanin in anti-melanogenic activity by using human melanoma MNT1 cells. They demonstrated that cajanin down-regulated melanin formation by decreased tyrosinase, TRP-1 and Dct expression. The study can be improved with the following specific points being appropriately addressed.
- In figure 2B, the authors showed that cajanin suppressed the cell proliferation over time. Do you think it is related to the color of MNT1 culture medium in figure 3A. Also, the authors did not discuss the effect of cajanin dosage in cell dividing. Is the cajanin dosage (50 μM) safe for pharmacotherapy?
- In figure 3E, please explain the X-axis labeling with log [cajanin] uM. Did they show significant differences between each other?
- In figure 4B, the authors measured tyrosinase activity with cajanin treatment in a cell-free system. Did authors have a positive control, such as adding tyrosinase, to demonstrate that the OD readings corresponded with the dopachrome formation?
- In figure 6A and 7A, the control group showed decreased MITF over time. It indicated that cajanin played a critical role at early stage. The author should discuss the observation carefully. It is meaningful to include shorter time points, such as 4,8,12 hours, to check the effect of cajanin.
Author Response
Response to reviewer
Reviewer 1
The authors investigated the molecular mechanism of cajanin in anti-melanogenic activity by using human melanoma MNT1 cells. They demonstrated that cajanin down-regulated melanin formation by decreased tyrosinase, TRP-1 and Dct expression. The study can be improved with the following specific points being appropriately addressed.
- In figure 2B, the authors showed that cajanin suppressed the cell proliferation over time. Do you think it is related to the color of MNT1 culture medium in figure 3A. Also, the authors did not discuss the effect of cajanin dosage in cell dividing. Is the cajanin dosage (50 μM) safe for pharmacotherapy?
Response: The color of melanin content extracted from melanocytes can be influenced by the cell number. To minimize to misinterpretation of anti-melanogenic activity, the melanin content was evaluated through both the direct observation using photographs (Figure 3a and b) and the ratio between melanin optical density (% µg melanin) and total protein content (µg) extracted from melanocytes (Figure 3c and d).
Revision:
Results on Page 3, Line 114-116: “The ratio between the cellular melanin and total protein content extracted from MNT1 cells was represented as a relative value to untreated control cells to minimize the interference from antiproliferative effect of cajanin.”
Discussion on Page 8-9, Line 239-246: “The overproduction of melanin content in various hyperpigmented disorders, including melasma and melanoma is involved with the up-regulation of melanogenesis pathway and hyperproliferation of melanocytes [26-28]. Interestingly, the expression of MITF also mediates proliferation and differentiation in melanocytes [29]. It has been reported that natural compound effectively diminished melanin content in melanocytes through suppression of cell proliferation [30]. The antiproliferative effect and down-regulation MITF related proteins of cajanin (50 μM) strongly support the further development of this isoflavonoid as an effective anti-melanogenic compound.”
References on Page 14-15, Line 514-524 added:
- Suzuki, I.; Kato, T.; Motokawa, T.; Tomita, Y.; Nakamura, E.; Katagiri, T. Increase of pro-opiomelanocortin mRNA prior to tyrosinase, tyrosinase-related protein 1, dopachrome tautomerase, Pmel-17/gp100, and P-protein mRNA in human skin after ultraviolet B irradiation. J Invest Dermatol 2002, 118, 73-78. DOI: 10.1046/j.1523-1747.2002.01647.x
- Young Kang, H.; Ortonne, J.P. Melasma update. Actas Dermosifiliogr 2009, 100 Suppl 2, 110-113. DOI: 10.1016/s0001-7310(09)73385-x
- Haass, N.K.; Herlyn, M. Normal human melanocyte homeostasis as a paradigm for understanding melanoma. J Investig Dermatol Symp Proc 2005, 10, 153-63. DOI: 10.1111/j.1087-0024.2005.200407.x
- Goding, C.; Meyskens, F.L. Jr. Microphthalmic-associated transcription factor integrates melanocyte biology and melanoma progression. Clin Cancer Res 2006, 12,1069-1073. DOI: 10.1158/1078-0432.CCR-05-2648
- Kang, W.; Choi, D.; Park, S.; Park, T. Carvone decreases melanin content by inhibiting melanoma cell proliferation via the cyclic adenosine monophosphate (cAMP) pathway. Molecules 2020, 25, 5191. DOI: 10.3390/molecules25215191
- In figure 3E, please explain the X-axis labeling with log [cajanin] uM. Did they show significant differences between each other?
Response: To demonstrate the dose-response curve for IC50 determination via nonlinear regression, the concentration is normally presented as logarithmic scale.
Revision:
Results on Page 3, Line 125-127: “The relationship between cellular melanin content and concentration in logarithmic (log) scale was generated for determining the half maximum inhibitory concentration (IC50) of cajanin in human melanin-producing cells.”
Moreover, the significant sign was added in the figure 3e of the revised manuscript as suggested by reviewer.
- In figure 4B, the authors measured tyrosinase activity with cajanin treatment in a cell-free system. Did authors have a positive control, such as adding tyrosinase, to demonstrate that the OD readings corresponded with the dopachrome formation?
Response: This study aimed to investigate the effect of cajanin on melanogenic activity in human melanin-producing cells. Therefore, the direct inhibition of melanogenic enzyme, especially tyrosinase was evaluated based on cellular proteins isolated from MNT1 cells. The optical density (OD) of formed dopachrome after adding l-DOPA in MNT1 protein lysate containing tyrosinase was determine at 490 nm according to Truong, et al. (2017). Moreover, all OD values was subtracted with OD of blank control that composed of l-DOPA and cajanin prior calculating of %Tyrosinase activity to minimize the interference. To confirm the determination of tyrosinase activity in cell-free tyrosinase activity assay, the results of mushroom tyrosinase which was used as a positive control was added in the figure 4b and
Revision:
Results on Page 5, Line 156-157: “Notably, mushroom tyrosinase (100 unit/mL) was used as a positive control to confirm to conversion of l-DOPA to dopachrome in cell-free tyrosinase activity assay.”
References on Page 16, Line 567-568 added:
- Truong, X.T.; Park, S.H.; Lee, Y.G.; Jeong, H.Y.; Moon, J.H.; Jeon, T.I. Protocatechuic acid from pear inhibits melanogenesis in melanoma cells. Int J Mol Sci 2017, 18, 1809. DOI:10.3390/ijms18081809
- In figure 6A and 7A, the control group showed decreased MITF over time. It indicated that cajanin played a critical role at early stage. The author should discuss the observation carefully. It is meaningful to include shorter time points, such as 4,8,12 hours, to check the effect of cajanin.
Response: We appreciated this valuable comment from the reviewer. We agree with the reviewer that the investigation on the related underlying mechanism at 4,8,12 h would bring more information about anti-melanogenic activity of cajanin in human melanin-producing cells. However, Newton et al, (2007) revealed that there was no significant alteration of mRNA and protein levels of MITF in human melanocyte during 3-24 h of the incubation time. Based on the significant reduction of melanin content in MNT1 cells after treatment with 50 µM cajanin for 24 h, the alteration of melanogenic proteins was primarily evaluated at this time point. At 24 h of incubation, the significant alteration of upstream regulating proteins, including pCREB/CREB and MITF as well as tyrosinase corresponded with the diminution of melanin content in cajanin-treated MNT1 cells. Additionally, at the later time point (48-72 h), the reduced expression level of tyrosinase, TPC1 and Dct was correlated with the prolong anti-melanogenic effect of cajanin. These data strongly suggest that cajanin mediates anti-melanogenic activity via modulation on MITF-related proteins.
Revision:
Discussion on Page 9, Line 287-293: “These results evidently suggested that cajanin might inhibit MITF expression through suppression on gene expression and stimulation of protein degradation consequence with diminished expression of tyrosinase family proteins. Although it has been revealed that there is no significant alteration of MITF during 3-24 h of culturing human melanocytes [44], the modulation of upstream molecules, including pCREB/CREB, MITF and pERK/ERK might be early observed before the reduction of downstream tyrosinase in cajanin-treated MNT1 cells.”
References on Page 15, Line 559-560 added:
- Newton, R.A.; Cook, A.L.; Roberts, D.W.; Leonard, J.H.; Sturm, R.A. Post-transcriptional regulation of melanin biosynthetic enzymes by cAMP and resveratrol in human melanocytes. J Invest Dermatol 2007, 127, 2216-2227. DOI: 10.1038/sj.jid.5700840
Minor changes and re-formatting
Minor corrections on grammar have been made.
*All changes in the revised manuscript are marked in red color.

Reviewer 2 Report
The authors investigate the antimelanogenic effects of the isoflavanoid cajanin. It is previously described in the literature that this substance inhibits the mushroom tyrosinase enzyme. The authors of this manuscript then set out to investigate the activity of this substance on human tyrosinase. In addition to the inhibition tests with human tyrosinase, the authors also present the results of cell viability and proliferation assay tests, determination of cellular melanin content, cell-free tyrosinase activity assay, determination of mRNA expression level by RT-qPCR, Western blot analysis in addition to statistical analyses.
# line 68: cajanin is a isoflavonoid (isoflavone).
# line 228: A 2018 reference (Promden et al.) is cited showing that the isoflavonoid cajanin has inhibitory effects on mushroom tyrosinase. Are there any significant structural differences described between human and mushroom tyrosinases that could explain the differences in the results? This article by Mayr et al. (J. Nat. Prod. 2019, 82, 1, 136–147) provides critical opinion on tyrosinase inhibition tests and the expected results.
Author Response
Response to reviewer
Reviewer 2
The authors investigate the antimelanogenic effects of the isoflavanoid cajanin. It is previously described in the literature that this substance inhibits the mushroom tyrosinase enzyme. The authors of this manuscript then set out to investigate the activity of this substance on human tyrosinase. In addition to the inhibition tests with human tyrosinase, the authors also present the results of cell viability and proliferation assay tests, determination of cellular melanin content, cell-free tyrosinase activity assay, determination of mRNA expression level by RT-qPCR, Western blot analysis in addition to statistical analyses.
- line 68: cajanin is a isoflavonoid (isoflavone).
Response: We would like to thank the reviewer for pointing this mistake.
Revision:
Introduction on Page 2, Line 67-69: “Cajanin, an isoflavonoid isolated from heartwood of Dalbergia parviflora Roxb. (Leguminosae) has been previously reported to have potent inhibitory effect on enzymatic activity of tyrosinase isolated from mushroom and murine sources [17].”
References on Page 14, Line 493-495 added:
- Promden, W.; Viriyabancha, W.; Monthakantirat, O.; Umehara, K.; Noguchi, H.; De-Eknamkul, W. Correlation between the potency of flavonoids on mushroom tyrosinase inhibitory activity and melanin synthesis in melanocytes. Molecules 2018, 23, 1403. DOI:10.3390/molecules23061403
- line 228: A 2018 reference (Promden et al.) is cited showing that the isoflavonoid cajanin has inhibitory effects on mushroom tyrosinase. Are there any significant structural differences described between human and mushroom tyrosinases that could explain the differences in the results? This article by Mayr et al. (J. Nat. Prod. 2019, 82, 1, 136–147) provides critical opinion on tyrosinase inhibition tests and the expected results.
Revision
Discussion on Page 9, Line 252-256: “The unique structure and amino sequence, including lack of thioether bond closely at the active site and containing EGF domain at the intra-melanosomal region of human tyrosinase cloud be the critical factors involving with the deficient activity against human tyrosinase of various mushroom tyrosinase inhibitors [33,34].”
References on Page 15, Line 530-534 added:
- Lai, X.; Soler-Lopez, M.; Wichers, H.J.; Dijkstra, B.W. Large-scale recombinant expression and purification of human tyrosinase suitable for structural studies. PLoS One 2016, 11, e0161697. DOI: 10.1371/journal.pone.0161697
- Mann, T.; Gerwat, W.; Batzer, J.; Eggers, K.; Scherner, C.; Wenck, H.; Stäb, F.; Hearing, V.J.; Röhm, K.H.; Kolbe, L. Inhibition of human tyrosinase requires molecular motifs distinctively different from mushroom tyrosinase. J Invest Dermatol 2018, 138, 1601-1608. DOI: 10.1016/j.jid.2018.01.019
Minor changes and re-formatting
Minor corrections on grammar have been made.
*All changes in the revised manuscript are marked in red color.

Reviewer 3 Report
The present manuscript describes the effect of non-toxic doses of cajanin on melanin-producing MNT1 cells. It also describes that the inhibitory effect of cajanin on the production of melanin in this type of cells is through the regulation of the CREB and ERK signaling pathways, decreasing the expression of MITF transcriptional factor, which in turn decreases the expression of the melanin synthesis enzymes TYR and Dct. All these data support the potential of cajanin as an anti-pigment agent with a safe use in the treatment of patients with hyperpigmentation disorders.
In my opinion, the manuscript is writhed adequately and presents strong evidence of the potential claimed of cajanin in this class of melanin-producing cells.
However, I consider that the following suggestions, could slightly improve the manuscript:
I consider that the paragraph about Fig 2B, page 2, lines 84-86, should be changed to “Figure 2B demonstrates that there was a clear increase in the % of proliferation after culturing MNT1 cells at low density (2 × 103 cells/well in 96-well plates) which was not repressed by 0-10 μM of cajanin for 48-72 h.”
In the text, the letters of each figure are in uppercase, meanwhile in the figure legends the letters are in lowercase…
Author Response
Response to reviewer
Reviewer 3
The present manuscript describes the effect of non-toxic doses of cajanin on melanin-producing MNT1 cells. It also describes that the inhibitory effect of cajanin on the production of melanin in this type of cells is through the regulation of the CREB and ERK signaling pathways, decreasing the expression of MITF transcriptional factor, which in turn decreases the expression of the melanin synthesis enzymes TYR and Dct. All these data support the potential of cajanin as an anti-pigment agent with a safe use in the treatment of patients with hyperpigmentation disorders.
In my opinion, the manuscript is writhed adequately and presents strong evidence of the potential claimed of cajanin in this class of melanin-producing cells.
However, I consider that the following suggestions, could slightly improve the manuscript:
- I consider that the paragraph about Fig 2B, page 2, lines 84-86, should be changed to “Figure 2B demonstrates that there was a clear increase in the % of proliferation after culturing MNT1 cells at low density (2 × 103 cells/well in 96-well plates) which was not repressed by 0-10 μM of cajanin for 48-72 h.”
Response: We would like to thank the reviewer for the suggestion.
Revision:
Results on Page 2-3, Line 91-93: Figure 2b demonstrates that there was a clear increase in the % of proliferation after culturing MNT1 cells at low density (2 × 103 cells/well in 96-well plates) which was not re-pressed by 0-10 μM of cajanin for 48-72 h.
- In the text, the letters of each figure are in uppercase, meanwhile in the figure legends the letters are in lowercase…
Response: All letters of each figure were changed to lowercase in figures, figure legends and main text.
Minor changes and re-formatting
Minor corrections on grammar have been made.
*All changes in the revised manuscript are marked in red color.

Reviewer 4 Report
This article by P. Netcharoensirisuk shows that Cajanin, an isoflavonoid derived from Dalbergia parviflora Roxb., has a potential to inhibit melanin production in cultured human cell line MNT1, and describes its molecular mechanism of action. In the study on the mechanism of action, the expression of genes involved in melanin production has been investigated in mRNA and protein levels. This article is of interest for researchers working on cosmetic and health sciences. However, this article does not appear to contain sufficient experimental results and explanations to support the authors’ conclusion. Since other flavonoids and natural compounds have already been reported to inhibit melanin production with a similar molecular mechanism (ref. 6, 26, and 27), this study seems to be significant because of the addition of Cajanin to these compounds with anti-melanogenic activity. Therefore, this article needs to be improved for publication in Molecules. Details are given below.
Major points:
- The authors should indicate the time period during which MNT1 cells consistently produce melanin under the experimental conditions used in the study. Do MNT1 cells consistently produce melanin during the 72 h of the experiment (Fig. 3)? This question arises from the results that the protein level of MITF and pCREB have remarkably decreased over time (from time = 24 h to time = 72 h under non-treatment ‘control’ condition; Fig. 6A and Fig. 7A) as well as the result shown in Fig. 3D. Specifically, in the experiment shown Fig. 3B and 3D, the authors need to add at least time=0 h along with more measurement time point in detail. As the authors know, in order to investigate the mechanism of action of Cajanin in inhibiting melanin production, it is important to study the effects of Cajanin under a condition that MNT1 cells consistently produce melanin.
- Is the molecular model in Fig. 8 consistent with the findings from this study? This question arises from the following sets of the experimental results; (1) In the absence of Cajanin treatment, the protein levels of MITF and pCREB decreased over time (Fig. 6A and 7A). In contrast, the mRNA and protein levels of TYR, TRP1, and Dct were unchanged (Fig. 5B–D). (2) The expression level of pCREB is repressed by Cajanin only at time =24 h (Fig. 7A). In contrast, the level of MITF mRNA is significantly repressed by Cajanin at not only t = 24 h but also t = 48 h and 72 h (Fig. 5A). (3) In the presence of Cajanin, the protein level of MITF was significantly decreased at t = 24 h (Fig. 6A). TYR mRNA level was also significantly decreased at t = 24 h (Fig. 5B). In contrast, the mRNA levels of TRP-1 and Dct genes were not affected by Cajanin at t = 24 h (Fig. 5C and 5D). In my opinion, it is preferable to revise the molecular model or add a detailed description.
- Is the CREB or ERK pathway of primary importance in inhibition of melanin production by Cajanin? At the data point t = 24 h, it can be determined that Cajanin significantly affects the CREB pathway but have no significant effect on the ERK pathway (Fig. 7). To answer this question, it would be better to examine the efficacy of Cajanin for each pathway at various concentrations as well as various time points.
- It would be better to revise the Introduction and Discussion sections. The Discussion repeats what is described in the Introduction. Also, it seems that some of the information that should be in the introduction is in the Discussion.
Minor points:
- It would be better to describe the purity of Cajanin used in this study, in the Materials and methods.
- The efficacy of Cajanin in inhibiting melanin production at concentrations that do not affect the viability of quiescent and proliferating cells should be explained. In the case of Cajanin, a concentration of 50 μM is not a non-toxic concentration (Introduction, Fig. 2, and Conclusion).
- It would be better to describe the doubling time of MNT1 cells. In the experiment in Fig. 2B, were the cells in the control (no treatment) proliferating logarithmically during the 72 h of culture?
Author Response
Response to reviewer
Reviewer 4
This article by P. Netcharoensirisuk shows that Cajanin, an isoflavonoid derived from Dalbergia parviflora Roxb., has a potential to inhibit melanin production in cultured human cell line MNT1, and describes its molecular mechanism of action. In the study on the mechanism of action, the expression of genes involved in melanin production has been investigated in mRNA and protein levels. This article is of interest for researchers working on cosmetic and health sciences. However, this article does not appear to contain sufficient experimental results and explanations to support the authors’ conclusion. Since other flavonoids and natural compounds have already been reported to inhibit melanin production with a similar molecular mechanism (ref. 6, 26, and 27), this study seems to be significant because of the addition of Cajanin to these compounds with anti-melanogenic activity. Therefore, this article needs to be improved for publication in Molecules. Details are given below.
Major points:
- The authors should indicate the time period during which MNT1 cells consistently produce melanin under the experimental conditions used in the study. Do MNT1 cells consistently produce melanin during the 72 h of the experiment (Fig. 3)? This question arises from the results that the protein level of MITF and pCREB have remarkably decreased over time (from time = 24 h to time = 72 h under non-treatment ‘control’ condition; Fig. 6A and Fig. 7A) as well as the result shown in Fig. 3D. Specifically, in the experiment shown Fig. 3B and 3D, the authors need to add at least time=0 h along with more measurement time point in detail. As the authors know, in order to investigate the mechanism of action of Cajanin in inhibiting melanin production, it is important to study the effects of Cajanin under a condition that MNT1 cells consistently produce melanin.
Response: The increment of melanin content in MNT1 cells after culture for 24-72 h presented in this study corresponds with the previous study reported about the logarithmical growth phase of MNT1 during 24-72 h when melanogenic proteins are activated. In accordance with the melanin result of control group in figure 3d, there was no significant alteration of melanogenic enzymes, including tyrosinase, TRP-1 and Dct during 24-72 h of culturing MNT1 cells (Figure 5 and 6). The short half-life (approximately 0.5-2.5 h) of MITF and negative feedback mechanism might influence on the gradual reduction of MITF and pCREB during 24-72 h of the incubation time, while the expression of tyrosinase family proteins is sustainable.
Revision:
Discussion on Page 9, Line 262-274: “Human melanoma MNT1 cells are highly pigmented cells that have been widely used for investigating melanogenic activity [19,35]. The increment of cellular melanin content (Figure 3b and d) associating with the sustainable level of melanogenic enzymes, including tyrosinase, TRP-1 and Dct (Figure 6b, c and d) evidenced the melanogenesis in MNT1 cells during 24-72 h of the incubation time performed in this study. Additionally, MNT1 cells possess the logarithmical growth after culture for 24-72 h in which melanogenic proteins are activated [18]. It should be noted that the gradually decreased MITF expression level after the incubation of MNT1 cells for 24-72 h could be the result of short half-life (approximately 0.5-2.5 h) of MITF [36] and negative feedback mechanism from downstream proteins [37]. Intriguingly, the expression of tyrosinase family proteins in MNT1 cells did not altered correspondingly with the reduction of MITF. It has been reported that tyrosinase is rapidly transferred into melanosome for preventing proteasome degradation and initiating melanin production in hyperpigmented cells [38].”
References on Page 14, Line 496-501 and Page 15, Line 535-546 added:
- Hoek, K.; Rimm, D.L.; Williams, K.R.; Zhao, H.; Ariyan, S.; Lin, A.; Kluger, H.M.; Berger, A.J.; Cheng, E.; Trombetta, E.S.; Wu, T.; Niinobe, M.; Yoshikawa, K.; Hannigan, G.E.; Halaban, R. Expression profiling reveals novel pathways in the transformation of melanocytes to melanomas. Cancer Res 2004, 64, 5270-5282. DOI: 10.1158/0008-5472.CAN-04-0731
- KleszczyÅ„ski, K.: Kim, T.K.; Bilska, B.; Sarna, M.; Mokrzynski, K.; Stegemann, A.; Pyza, E.; Reiter, R.J.; Steinbrink, K.; Böhm, M.; Slominski, A.T. Melatonin exerts oncostatic capacity and decreases melanogenesis in human MNT-1 melanoma cells. J Pineal Res 2019, 67, e12610. DOI: 10.1111/jpi.12610
- Netcharoensirisuk, P.; Abrahamian, C.; Tang, R.; Chen, C.C.; Rosato, A.S.; Beyers, W.; Chao, Y.K.; Filippini, A.; Di Pietro, S.; Bartel, K.; Biel, M.; Vollmar, A.M.; Umehara, K.; De-Eknamkul, W.; Grimm, C. Flavonoids increase melanin production and reduce proliferation, migration and invasion of melanoma cells by blocking endolysosomal/melanosomal TPC2. Sci Rep 2021, 11, 8515. DOI: 10.1038/s41598-021-88196-6
- Wellbrock, C.; Arozarena, I. Microphthalmia-associated transcription factor in melanoma development and MAP-kinase pathway targeted therapy. Pigment Cell Melanoma Res 2015, 28, 390-406. DOI: 10.1111/pcmr.12370
- Lin, C.B.; Babiarz, L.; Liebel, F.; Roydon Price, E.; Kizoulis, M.; Gendimenico, G.J.; Fisher, D.E.; Seiberg, M. Modulation of microphthalmia-associated transcription factor gene expression alters skin pigmentation. J Invest Dermatol 2002, 119, 1330-40. DOI: 10.1046/j.1523-1747.2002.19615.x
- Watabe, H.; Valencia, J.C.; Yasumoto, K.; Kushimoto, T.; Ando, H.; Muller, J.; Vieira, W.D.; Mizoguchi, M.; Appella, E.; Hearing, V.J. Regulation of tyrosinase processing and trafficking by organellar pH and by proteasome activity. J Biol Chem 2004, 279, 7971-7981. DOI: 10.1074/jbc.M309714200
- Is the molecular model in Fig. 8 consistent with the findings from this study? This question arises from the following sets of the experimental results; (1) In the absence of Cajanin treatment, the protein levels of MITF and pCREB decreased over time (Fig. 6A and 7A). In contrast, the mRNA and protein levels of TYR, TRP1, and Dct were unchanged (Fig. 5B–D). (2) The expression level of pCREB is repressed by Cajanin only at time =24 h (Fig. 7A). In contrast, the level of MITF mRNA is significantly repressed by Cajanin at not only t = 24 h but also t = 48 h and 72 h (Fig. 5A). (3) In the presence of Cajanin, the protein level of MITF was significantly decreased at t = 24 h (Fig. 6A). TYR mRNA level was also significantly decreased at t = 24 h (Fig. 5B). In contrast, the mRNA levels of TRP-1 and Dct genes were not affected by Cajanin at t = 24 h (Fig. 5C and 5D). In my opinion, it is preferable to revise the molecular model or add a detailed description.
Response: Schematic diagram presenting a proposed anti-melanogenic activity of cajanin in Figure 8 was corrected by presenting only the regulatory proteins investigated in this study. Moreover, the description of proposed underlying mechanism was added in the legend of Figure 8 of revised manuscript
Revision:
Conclusions on Page 12-13, Line 427-435: as “Figure 8. Schematic diagram presenting a proposed anti-melanogenic activity of cajanin in human melanin-producing cells. Cajanin modulates MITF, a melanogenic transcription factor, at both pre- and post-translation steps through down-regulation of pCREB/CREB and activation of pERK/ERK signal, respectively. (a) At the initial period of treatment (24-48 h), the inhibition of pCREB/CREB signaling corresponds with decreased mRNA and protein levels of MITF and tyrosinase (TYR) mediated by cajanin. (b) Interestingly, the activation of pERK/ERK might play a role on dramatic decreased of MITF protein which consequently attenuates the expression levels of melanin synthesis enzymes including tyrosinase (TYR), TRP-1 and Dct (TRP-2) at the latter time point (72 h). These modulations on melanin regulating proteins eventually results in the prolong decreased melanin content in cajanin-treated melanocytes. The red arrows and symbols indicate the effect of cajanin in human melanin-producing cells.”
- Is the CREB or ERK pathway of primary importance in inhibition of melanin production by Cajanin? At the data point t = 24 h, it can be determined that Cajanin significantly affects the CREB pathway but have no significant effect on the ERK pathway (Fig. 7). To answer this question, it would be better to examine the efficacy of Cajanin for each pathway at various concentrations as well as various time points.
Response: In this study, the significant reduction of melanin production was observed in MNT1 cells only after culture with cajanin at 50 µM for 24-72 h although the minor diminution of melanin content was also presented in MNT1 cells treated with 10 µM cajanin for 72 h. Therefore, the investigation on regulatory proteins was mainly focused in MNT1 cells after culture with cajanin at 50 µM for 24-72 h. We agree with the reviewer that the alteration of upstream regulatory proteins (pCREB/CREB, MITF and pERK/ERK) might be observed at the early time point before the modulation of downstream signals (Tyrosinase, TRP-1 and Dct). Based on the significant reduction of melanin content in MNT1 cells after treatment with 50 µM cajanin for 24 h, the alteration of related proteins was primarily evaluated at this time point. At 24 h of incubation, the significant alteration of upstream proteins, including pCREB/CREB and MITF, as well as tyrosinase corresponded with the diminution of melanin content in cajanin-treated MNT1 cells. Additionally, at the later time point (48-72 h), the reduced expression level of tyrosinase, TRP-1 and Dct was correlated with the prolong anti-melanogenic effect of cajanin. These data suggest that cajanin mediate anti-melanogenic activity via modulation on MITF-related proteins. Although the decreased expression level of pCREB/CREB was presented only at 24 h, the up-regulation of pERK/ERK that triggers MITF degradation was also corresponded with the dramatic reduction of MITF protein expression following with the diminished level of tyrosinase, TRP-1 and Dct at 72 h.
Revision:
Results on Page 5, Line 169-173: “To investigate whether the diminution of cellular melanin and tyrosinase activity in cajanin-treated human melanin-producing cells was associated with the alteration of key melanogenic enzymes, the expression levels of MITF, tyrosinase, TRP-1 and Dct were mainly examined in MNT1 cells cultured with the effective concentration (50 µM) of cajanin for 24-72 h through both RT-qPCR and western blot analysis.”
Discussion on Page 9, Line 287-293: “These results evidently suggested that cajanin might inhibit MITF expression through suppression on gene expression and stimulation of protein degradation consequence with diminished expression of tyrosinase family proteins. Although it has been revealed that there is no significant alteration of MITF during 3-24 h of culturing human melanocytes [44], the modulation of upstream molecules, including pCREB/CREB, MITF and pERK/ERK might be early observed before the reduction of downstream tyrosinase in cajanin-treated MNT1 cells.”
References on Page 15, Line559-560 added:
- Newton, R.A.; Cook, A.L.; Roberts, D.W.; Leonard, J.H.; Sturm, R.A. Post-transcriptional regulation of melanin biosynthetic enzymes by cAMP and resveratrol in human melanocytes. J Invest Dermatol 2007, 127, 2216-2227. DOI: 10.1038/sj.jid.5700840
- It would be better to revise the Introduction and Discussion sections. The Discussion repeats what is described in the Introduction. Also, it seems that some of the information that should be in the introduction is in the Discussion.
Response: The Discussion section of revised manuscript was corrected as suggested by the reviewer.
Revision:
Discussion on Page 8, Line 223-238: “It has been documented that a variety of chemical reagents, therapeutic treatments and several pathological conditions are involved with melanogenesis disorders, especially the overproduction of melanin [22]. As MITF stimulates the expression of melanogenic enzymes, including tyrosinase, TRP-1 and Dct, the activation of this transcription factor accordingly increases melanin synthesis in melanocytes [8,23]. Both genetic mutations and exogenous stimuli can alter MITF expression level, consequently leading to melano-genesis dysregulation and pigmentation disorders [24]. Additionally, MITF has also been proposed as a diagnostic marker for melanoma [25]. Not only suppression on down-stream melanogenic proteins in human melanoma cells but also lightening of skin tone was reported in a clinical study after treatment with a MITF-repressing agent [14]. Inter-estingly, natural flavonoids have been found to provide an anti-melanogenic effect through down-regulated MITF expression [6]. This finding correlates with results of this present study wherein the attenuated mRNA (Figure 5a) and protein levels of MITF (Figure 6a) associated with the reduction of cellular melanin content (Figure 3a-e) was re-vealed in human melanoma MNT1 cells after treatment with 50 μM cajanin, an isoflavo-noid isolated from D. parviflora, for 24-72 h.”
Comments on Minor Points:
- It would be better to describe the purity of Cajanin used in this study, in the Materials and methods.
Revision:
Materials and Methods on Page 10, Line 297-298: “Cajanin with approximately 98% w/w was isolated and identified from D. parviflora as previously reported [17,45,46].”
Reference on Page 14, Line 493-495 and Page 16, Line 561-566 added:
- Promden, W.; Viriyabancha, W.; Monthakantirat, O.; Umehara, K.; Noguchi, H.; De-Eknamkul, W. Correlation between the potency of flavonoids on mushroom tyrosinase inhibitory activity and melanin synthesis in melanocytes. Molecules 2018, 23, 1403. DOI:10.3390/molecules23061403
- Umehara, K.; Nemoto, K.; Kimijima, K.; Matsushita, A.; Terada, E.; Monthakantirat, O.; De-Eknamkul, W.; Miyase, T.; Warashina, T.; Degawa, M.; Noguchi, H. Estrogenic constituents of the heartwood of Dalbergia parviflora. Phytochemistry 2008, 69, 546-552. DOI:10.1016/j.phytochem.2007.07.011
- De-Eknamkul, W.; Umehara, K.; Monthakantirat, O.; Toth, R.; Frecer, V.; Knapic, L.; Braiuca, P.; Noguchi, H.; Miertus, S. QSAR study of natural estrogen-like isoflavonoids and diphenolics from Thai medicinal plants. J Mol Graph Model 2011, 29, 784-794. DOI:10.1016/j.jmgm.2011.01.001
- The efficacy of Cajanin in inhibiting melanin production at concentrations that do not affect the viability of quiescent and proliferating cells should be explained. In the case of Cajanin, a concentration of 50 μM is not a non-toxic concentration (Introduction, Fig. 2, and Conclusion).
Revision: The word of “non-toxic concentration” was deleted and explain as
Results on Page 3, Line 105-106: “The effect of 1-50 μM cajanin which did not alter %cell viability after 24-h treatment was further evaluated in proliferation assay”
Results on Page 3, Line 114-120: “The ratio between the cellular melanin and total protein content extracted from MNT1 cells was represented as a relative value to untreated control cells to minimize the interference from antiproliferative effect of cajanin. Figure 3c demonstrates the reduction of cellular melanin content observed after incubation with 10-50 μM cajanin for 72 h. Although the minor diminution of melanin production was presented in MNT1 cells cultured with cajanin at lower concentration (10 μM), the significant anti-melanogenic effect was indicated only in the treatment of 50 μM cajanin.”
- It would be better to describe the doubling time of MNT1 cells. In the experiment in Fig. 2B, were the cells in the control (no treatment) proliferating logarithmically during the 72 h of culture?
Revision:
Results on Page 2, Line 88-91: “Since the doubling time of human melanocytes is approximately 72 h [18], the effect of cajanin on the proliferation of human melanoma MNT1 cells was observed after the incubation period of 24-72 h that covers the logarithmic growth phase of the cells [19].”
References on Page 14, Line 496-501 added:
- Hoek, K.; Rimm, D.L.; Williams, K.R.; Zhao, H.; Ariyan, S.; Lin, A.; Kluger, H.M.; Berger, A.J.; Cheng, E.; Trombetta, E.S.; Wu, T.; Niinobe, M.; Yoshikawa, K.; Hannigan, G.E.; Halaban, R. Expression profiling reveals novel pathways in the transformation of melanocytes to melanomas. Cancer Res 2004, 64, 5270-5282. DOI: 10.1158/0008-5472.CAN-04-0731
- KleszczyÅ„ski, K.: Kim, T.K.; Bilska, B.; Sarna, M.; Mokrzynski, K.; Stegemann, A.; Pyza, E.; Reiter, R.J.; Steinbrink, K.; Böhm, M.; Slominski, A.T. Melatonin exerts oncostatic capacity and decreases melanogenesis in human MNT-1 melanoma cells. J Pineal Res 2019, 67, e12610. DOI: 10.1111/jpi.12610
Minor changes and re-formatting
Minor corrections on grammar have been made.
*All changes in the revised manuscript are marked in red color.

Round 2
Reviewer 4 Report
The revised article has been improved, in response to the reviewers’ comments. In addition of the addition of the control sample in Fig. 4B, a detailed explanation and interpretation of the results were notably added. However, several points have not been addressed adequately. The article can be further improved by appropriately addressing the following points.
Major points:
- In relation to the previous comment #1 (regarding MNT1 cells producing melanin at a constant rate during the experimental period of 72 h under the authors’ experimental condition): As Authors answered, the protein levels of TYR, TRP1 and Dct did not decrease during the experimental period (t=24~72 h; Fig. 6B–D). This implies that MNT1 cells consistently produce melanin during the period. As shown in Fig. 3d, however, the amounts of melanin in the control and DMSO samples remained almost unchanged during the second 24 h (t=48~72 h), whereas those increased during the first 24 h (t=24~48 h). Therefore, the melanin amount (% μg melanin/μg protein) of the control and DMSO samples at t=0 h should be presented. This will show the increase in the melanin amount during the “first” 24 h (t=0~24) and will make it clear that MNT1 cells are consistently producing melanin during the experimental period of 72 h (t=0~72 h) or not. As Author understand, the status of melanin production in MNT1 cells during the experimental period is important to understand the results of the subsequent experiments shown in Figs. 4–7.
- In relation to the previous comment #3; lines 290–292 (regarding the response of the CREB and ERK signal pathways to the cajanin treatment): Authors mention that the modulation of pERK/ERK might be early observed before the reduction of tyrosinase in the cajanin-treated MNT1 cells. However, in the results (Figs. 6B and 7B), pERK/ERK is significantly unchanged by cajanin at t=24 h, whereas the protein level of TYR was decreased by cajanin at the same time point.
- In relation to the comment #1 above; Figs. 2B, 3B (3D), 4A, 5, 6, 7: Experiments that investigate changes in cells over time require to show the results of the control samples at the time point t=0 h.
Minor points
- In relation to the previous minor comment #3; Fig. 2B and line 88: Please note the doubling time of MNT1 cells under the Authors’ experimental condition. Judging from the results in Fig. 2B, it does not appear that the doubling time of MNT1 cells is 72 h.
- 4A and lines 361–362: Please add an explanation/interpretation for the increase in cellular TYR activity in the untreated MNT1 cells over time, even though each cell lysate used in the experiments contains the same amount of protein (100 μg). Due to this increase in control TRY activity, it appears that 50 μM of cajanin significantly inhibits the cellular TRY activity, especially at t=72 h. This comment may be related to the comment #1 described above.
Author Response
Response to reviewer
Reviewer 4
The revised article has been improved, in response to the reviewers’ comments. In addition of the addition of the control sample in Fig. 4B, a detailed explanation and interpretation of the results were notably added. However, several points have not been addressed adequately. The article can be further improved by appropriately addressing the following points.
Major points:
- In relation to the previous comment #1 (regarding MNT1 cells producing melanin at a constant rate during the experimental period of 72 h under the authors’ experimental condition): As Authors answered, the protein levels of TYR, TRP1 and Dct did not decrease during the experimental period (t=24~72 h; Fig. 6B–D). This implies that MNT1 cells consistently produce melanin during the period. As shown in Fig. 3d, however, the amounts of melanin in the control and DMSO samples remained almost unchanged during the second 24 h (t=48~72 h), whereas those increased during the first 24 h (t=24~48 h). Therefore, the melanin amount (% μg melanin/μg protein) of the control and DMSO samples at t=0 h should be presented. This will show the increase in the melanin amount during the “first” 24 h (t=0~24) and will make it clear that MNT1 cells are consistently producing melanin during the experimental period of 72 h (t=0~72 h) or not. As Author understand, the status of melanin production in MNT1 cells during the experimental period is important to understand the results of the subsequent experiments shown in Figs. 4–7.
Response: To respond to the major and minor points of Reviewer 4’s comments, the additional experiments were conducted to observe the changes of various parameters during the first 24 h. All results were shown in the supplementary materials (Figure S1-S6). For the melanin content in MNT1 cells, it was found that there was a slight increase of melanin content at 12 h of the incubation time (Figure S2).
Revision:
Results on Page 3, Line 113-115: “It should be noted that the gradually augmented melanin content in human MNT1 cells was early observed at 12 h of the incubation time (Supplementary data; Figure S2).”
Discussion on Page 9, Line 267-271: “The increment of cellular melanin content (Figure 3b, 3d and Supplementary data; Figure S2) associating with the sustainable level of melanogenic enzymes, including tyrosinase, TRP-1 and Dct (Figure 6b, 6c, 6d and Supplementary data; Figure S4) evidenced the melanogenesis in MNT1 cells during 0-72 h of the incubation time performed in this study.”
Supplementary Materials on Page 3, Figure S2: “Figure S2. The melanin production in MNT1 cells after culture for 0-24 h. (a) Treatment for 0-12 h with neither 50 μM cajanin nor 20 µM tyrosinase inhibitor (TYRi; 4-butylresorcinol) altered melanin production in human MNT1 cells. (b) During the incubation period of 0-24 h, the gradual increase of melanin content was early noted in human melanin-producing MNT1 cells since 12 h. DMSO was used as a vehicle control. Experiments were performed in triplicate and data are expressed as mean ± SEM. ***p < 0.005 versus non-treated cells (Ctrl).”
- In relation to the previous comment #3; lines 290–292 (regarding the response of the CREB and ERK signal pathways to the cajanin treatment): Authors mention that the modulation of pERK/ERK might be early observed before the reduction of tyrosinase in the cajanin-treated MNT1 cells. However, in the results (Figs. 6B and 7B), pERK/ERK is significantly unchanged by cajanin at t=24 h, whereas the protein level of TYR was decreased by cajanin at the same time point.
Response: The expression levels of signaling molecules mediating MITF expression in human MNT1 cells cultured with cajanin for 0-12 h was provided in Figure S6 of the supplementary material. Western blot analysis demonstrated no significant alteration of (a) pCREB/CREB and (b) pERK/ERK levels in MNT cells treated with 50 μM of cajanin for 0-12 h compared with the control cells.
Revision:
Discussion on Page 9-10, Line 301-305: “Correspondence with the previous report about no significant alteration of MITF during the initial period (approximately at 3-24 h) of culturing human melanocytes [44], the modulation of upstream molecules, including pCREB/CREB and MITF were also not observed in MNT1 cells either in the presence or absence of cajanin for 0-12 h (Supplementary data; Figure S4-S6).”
Supplementary Materials on Page 7, Figure S6: “Figure S6. The expression levels of signaling molecules mediating MITF expression in human MNT1 cells cultured with cajanin for 0-12 h. Western blot analysis demonstrated that there was no significant alteration of (a) pCREB/CREB and (b) pERK/ERK levels in MNT cells treated with 50 μM of cajanin for 0-12 h compared with the control cells. Experiments were performed in triplicate and data are expressed as mean ± SEM.”
Reference was added on Page 16, Line 577-578
- Newton, R.A.; Cook, A.L.; Roberts, D.W.; Leonard, J.H.; Sturm, R.A. Post-transcriptional regulation of melanin biosynthetic enzymes by cAMP and resveratrol in human melanocytes. J Invest Dermatol 2007, 127, 2216-2227. DOI: 10.1038/sj.jid.5700840
- In relation to the comment #1 above; Figs. 2B, 3B (3D), 4A, 5, 6, 7: Experiments that investigate changes in cells over time require to show the results of the control samples at the time point t=0 h.
Response: The effects of cajanin treatment for 0-12 h in MNT1 cells were provided in the supplementary data. Nevertheless, the antiproliferative effect of cajanin was not evaluated in MNT1 cells during 0-12 h of the incubation time which is earlier than the doubling time of 13.75 h.
Revision:
Supplementary Materials on Page 2, Figure S1: Figure S1. Growth curve of MNT1 cells. Human melanoma MNT1 cells were seed at density of 1 × 105 cells/well into 6-well plates and further cultured for 24-96 h. The number of cells at each time point were counted after staining with tryptan blue and generated the growth curve. Based on analysis via GraphPad Prism 9.0 software, the MNT1 cells under the present condition possess the doubling time at 13.75 h. Experiments were performed in triplicate and data are expressed as mean ± SEM.
Supplementary Materials on Page 3, Figure S2: Figure S2. The melanin production in MNT1 cells after culture for 0-24 h. (a) Treatment for 0-12 h with neither 50 μM cajanin nor 20 µM tyrosinase inhibitor (TYRi; 4-butylresorcinol) altered melanin production in human MNT1 cells. (b) During the incubation period of 0-24 h, the gradual increase of melanin content was early noted in human melanin-producing MNT1 cells since 12 h. DMSO was used as a vehicle control. Experiments were performed in triplicate and data are expressed as mean ± SEM. ***p < 0.005 versus non-treated cells (Ctrl).
Supplementary Materials on Page 4, Figure S3: Figure S3. Tyrosinase activity in MNT1 cells cultured for 0-12 h. Human MNT1 cells at density of 1 × 105 cells/well in 6-well plate were treated with 50 μM cajanin, 20 µM tyrosinase inhibitor (TYRi; 4-butylresorcinol) or DMSO (vehicle) for 0-12 h before being subjected for determining tyrosinase activity via cell-based assay. There was no significant alteration of tyrosinase activity in all treatment groups compared with the non-treated control cells at the same time point. Additionally, the minor increase of tyrosinase activity relatively to 0 h was observed in MNT1 cells cultured for 6-12 h. Experiments were performed in triplicate and data are expressed as mean ± SEM.
Supplementary Materials on Page 5, Figure S4: Figure S4. The mRNA expression levels of melanogenesis-related proteins in human MNT1 cells cultured with cajanin for 0-12 h. Real-time quantitative RT-PCR revealed no significant alteration of mRNA levels of (a) MITF, (b) tyrosinase (TYR), (c) TRP-1 and (d) Dct in MNT1 cells cultured with 50 μM cajanin for 0-12 h compared with non-treated control cells at the same time point. Although minor alteration of the mRNA levels was observed in MNT1 cells cultured for 6-12 h, there was not significantly different when compared with the cells at 0 h. Data obtained from qRT-PCR were normalized to GAPDH expression level and represented relative to control at 0 h. Experiments were performed in triplicate and data are expressed as mean ± SEM.
Supplementary Materials on Page 6, Figure S5: Figure S5. The expression levels of melanogenesis-related proteins in human MNT1 cells cultured with cajanin for 0-12 h. Western blot analysis demonstrated that there was no significant alteration of (a) MITF, (b) tyrosinase (TYR), (c) TRP-1 and (d) Dct protein levels in MNT1 cells treated with 50 μM of cajanin for 0-12 h compared with non-treated control cells. Experiments were performed in triplicate and data are expressed as mean ± SEM.
Supplementary Materials on Page 7, Figure S6: Figure S6. The expression levels of signaling molecules mediating MITF expression in human MNT1 cells cultured with cajanin for 0-12 h. Western blot analysis demonstrated that there was no significant alteration of (a) pCREB/CREB and (b) pERK/ERK levels in MNT1 cells treated with 50 μM of cajanin for 0-12 h compared with non-treated control cells. Experiments were performed in triplicate and data are expressed as mean ± SEM.
Minor points:
- In relation to the previous minor comment #3; Fig. 2B and line 88: Please note the doubling time of MNT1 cells under the Authors’ experimental condition. Judging from the results in Fig. 2B, it does not appear that the doubling time of MNT1 cells is 72 h.
Response: To identify the doubling time of MNT1 cells under the same condition of the evaluation of melanin content, the growth curve was generated from MNT1 cells seeded at density of 1 × 105 cells/well in 6-well plates and further incubated for 0-96 h. The growth curve as indicated in Figure S1 in supplementary material was analyzed via GraphPad Prism 9.0 software which demonstrated that MNT1 cells possess the doubling time of 13.75 h.
Revision:
Result on Page 2, Line 88-91: “Although 72 h had been reported as a doubling time of human melanocytes [18], the MNT1 cells cultured under the present condition was found to have the doubling time of 13.75 h (Supplementary data; Figure S1).”
Supplementary Materials on Page 2, Figure S1: Figure S1. Growth curve of MNT1 cells. Human melanoma MNT1 cells were seed at density of 1 × 105 cells/well into 6-well plates and further cultured for 24-96 h. The number of cells at each time point were counted after staining with tryptan blue and generated the growth curve. Based on analysis via GraphPad Prism 9.0 software, the MNT1 cells under the present condition possess the doubling time at 13.75 h. Experiments were performed in triplicate and data are expressed as mean ± SEM.
Reference was added on Page 14, Line 514-516
- Hoek, K.; Rimm, D.L.; Williams, K.R.; Zhao, H.; Ariyan, S.; Lin, A.; Kluger, H.M.; Berger, A.J.; Cheng, E.; Trombetta, E.S.; Wu, T.; Niinobe, M.; Yoshikawa, K.; Hannigan, G.E.; Halaban, R. Expression profiling reveals novel pathways in the transformation of melanocytes to mel-anomas. Cancer Res 2004, 64, 5270-5282. DOI: 10.1158/0008-5472.CAN-04-0731
- 4A and lines 361–362: Please add an explanation/interpretation for the increase in cellular TYR activity in the untreated MNT1 cells over time, even though each cell lysate used in the experiments contains the same amount of protein (100 μg). Due to this increase in control TRY activity, it appears that 50 μM of cajanin significantly inhibits the cellular TRY activity, especially at t=72 h. This comment may be related to the comment #1 described above.
Response: To clarify the increase of cellular tyrosinase activity in MNT1 after culturing over time, the tyrosinase activity in MNT1 cells cultured for 0-12 h was additionally determined and added in Figure S3 of the supplementary material.
Revision:
Discussion on Page 9, Line 279-285: “Notably, the maturation of melanosome from stage I to IV tightly regulates melanogenic activity of tyrosinase. Despite containing the activity during located in ribosome and rough endoplasmic reticulum, the catalytic function of tyrosinase is completely activated inside melanosome, especially at the latter stages of III and IV [1]. These might involve with the increased activity of tyrosinase observed in MNT1 cells during 6-72 h of the incubation time (Figure 4a and Supplementary data; Figure S3).”
Supplementary Materials on Page 4, Figure S3: Figure S3. Tyrosinase activity in MNT1 cells cultured for 0-12 h. Human MNT1 cells at density of 1 × 105 cells/well in 6-well plate were treated with 50 μM cajanin, 20 µM tyrosinase inhibitor (TYRi; 4-butylresorcinol) or DMSO (vehicle) for 0-12 h before being subjected for determining tyrosinase activity via cell-based assay. There was no significant alteration of tyrosinase activity in all treatment groups compared with the non-treated control cells at the same time point. Additionally, the minor increase of tyroisnase activity relatively to 0 h was observed in MNT1 cells cultured for 6-12 h. Experiments were performed in triplicate and data are expressed as mean ± SEM.
Reference was added on Page 13, Line 479-480
- Slominski, A.; Tobin D.J.; Shibahara S.; Wortsman J. Melanin pigmentation in mammalian skin and its hormonal regulation. Physiol Rev 2004, 84, 1155-1228. DOI:10.1152/physrev.00044.2003
Minor changes and re-formatting
Minor corrections on grammar have been made.
*All changes in the revised manuscript are marked in red color.
